# Glutamatergic drive along the septo-temporal axis of hippocampus boosts prelimbic oscillations in the neonatal mouse

Joachim Ahlbeck[†], Lingzhen Song, Mattia Chini, Sebastian H Bitzenhofer, Ileana L Hanganu-Opatz*

Developmental Neurophysiology, Institute of Neuroanatomy, University Medical Center Hamburg-Eppendorf, Hamburg, Germany

**Abstract** The long-range coupling within prefrontal-hippocampal networks that account for cognitive performance emerges early in life. The discontinuous hippocampal theta bursts have been proposed to drive the generation of neonatal prefrontal oscillations, yet the cellular substrate of these early interactions is still unresolved. Here, we selectively target optogenetic manipulation of glutamatergic projection neurons in the CA1 area of either dorsal or intermediate/ventral hippocampus at neonatal age to elucidate their contribution to the emergence of prefrontal oscillatory entrainment. We show that despite stronger theta and ripples power in dorsal hippocampus, the prefrontal cortex is mainly coupled with intermediate/ventral hippocampus by phase-locking of neuronal firing via dense direct axonal projections. Theta band-confined activation by light of pyramidal neurons in intermediate/ventral but not dorsal CA1 that were transfected by *in utero* electroporation with high-efficiency channelrhodopsin boosts prefrontal oscillations. Our data causally elucidate the cellular origin of the long-range coupling in the developing brain.
DOI: https://doi.org/10.7554/eLife.33158.001

*For correspondence:
hangop@zmnh.uni-hamburg.de

Present address: [†]Department of Neurophysiology and Pathophysiology, University Medical Center Hamburg-Eppendorf, Hamburg, Germany

Competing interests: The authors declare that no competing interests exist.

## Introduction

In the adult rodent brain, coordinated patterns of oscillatory activity code in a frequency-specific manner for sensory and cognitive performance. For example, learning and memory critically depend on oscillations within theta frequency band (4–12 Hz) that functionally couple the medial prefrontal cortex (PFC) and hippocampus (HP) (*Siapas and Wilson, 1998*; *Benchenane et al., 2010*; *Brincat and Miller, 2015*; *Backus et al., 2016*; *Eichenbaum, 2017*; *Wirt and Hyman, 2017*). These frequency-tuned brain states are present already during early development, long before the memory and attentional abilities have fully matured. They have been extensively characterized and categorized according to their spatial and temporal structure (*Lindemann et al., 2016*). Network oscillations during development have a highly discontinuous and fragmented structure with bursts of activity alternating with 'silent' periods (*Hanganu et al., 2006*; *Seelke and Blumberg, 2010*; *Shen and Colonnese, 2016*; *Luhmann and Khazipov, 2018*). The most common oscillatory pattern, spindle bursts, synchronizes large cortical and subcortical networks within theta-alpha frequency range. It is accompanied by slow delta waves as well as by faster discharges (beta and gamma oscillations) that account for local activation of circuits (*Brockmann et al., 2011*; *Yang et al., 2016*).

In the absence of direct behavioral correlates, a mechanistic understanding of oscillatory rhythms in the developing brain is currently lacking. In sensory systems, spindle bursts have been proposed to act as a template facilitating the formation of cortical maps (*Dupont et al., 2006*; *Hanganu et al., 2006*; *Tolner et al., 2012*), whereas early gamma oscillations seem to control the organization of

**eLife digest** When memories are stored, or mental tasks performed, different parts of the brain need to communicate with each other to process and extract information from the environment. For example, the communication between two brain areas called the hippocampus and the prefrontal cortex is essential for memory and attention. However, it is still unclear how these interactions are established when the brain develops.

Now, by looking at how the hippocampus and the prefrontal cortex 'work' together in newborn mouse pups, Ahlbeck et al. hope to understand how these brain areas start to connect. In particular, the groups of neurons that kick start the development of the circuits required for information processing need to be identified.

Recording the brains of the pups revealed that electrical activity in a particular sub-division of the hippocampus activated neurons in the prefrontal cortex. In fact, a specific population of neurons in this area was needed for the circuits in the prefrontal cortex to mature.

In further experiments, the neurons from this population in the hippocampus were manipulated so they could be artificially activated in the brain using light. When stimulated, these neurons generated electrical activity, which was then relayed through the neurons all the way to the prefrontal cortex. There, this signal triggered local neuronal circuits. Thanks to this activation, these circuits could 'wire' together, and start establishing the connections necessary for mental tasks or memory in adulthood.

The brain of the mouse pups used by Ahlbeck et al. was approximately in the same developmental state as the brain of human fetuses in the second or third trimester of pregnancy. These findings may therefore inform on how the hippocampus and the prefrontal cortex start connecting in humans. Problems in the way brain areas interact during early development could be partly responsible for certain neurodevelopmental disorders and mental illnesses, such as schizophrenia. Understanding these processes at the cellular level may therefore be the first step towards finding potential targets for treatment.

DOI: https://doi.org/10.7554/eLife.33158.002

thalamocortical topography (*Minlebaev et al., 2011*; *Khazipov et al., 2013*). In limbic systems dedicated to mnemonic and executive abilities, the knowledge on the relevance of early network oscillations is even sparser. Few lesion studies, yet without selectivity for specific activity patterns, suggested that prefrontal-hippocampal communication during development might be necessary for the maturation of episodic memory (*Krüger et al., 2012*). Temporal associations between the firing and synaptic discharges of individual neurons and network oscillations in different frequency bands gave first insights into the cellular substrate of coordinated activity in neonates. Whereas in sensory systems, endogenous activation of sensory periphery drives entrainment of local circuitry through gap junction coupling as well as glutamatergic and GABAergic transmission (*Dupont et al., 2006*; *Hanganu et al., 2006*; *Minlebaev et al., 2009*), in developing prefrontal-hippocampal networks, the excitatory drive from the HP has been proposed to activate a complex layer- and frequency-specific interplay in the PFC (*Brockmann et al., 2011*; *Bitzenhofer and Hanganu-Opatz, 2014*; *Bitzenhofer et al., 2015*).

While most of this correlative evidence put forward the relevance of early oscillations beyond a simple epiphenomenal signature of developing networks, direct evidence for their causal contribution to circuit maturation is still missing. This is mainly due to the absence of a causal interrogation of developing networks, similarly to the investigations done in adult ones. Only recently the methodological difficulties related to area-, layer- and cell type-specific manipulations at neonatal age have been overcome (*Bitzenhofer et al., 2017a*; *Bitzenhofer et al., 2017b*). By these means, the local neuronal interplay generating beta-gamma oscillations in the PFC has been elucidated. However, the long-range coupling causing the activation of local prefrontal circuits is still unresolved. We previously proposed that the hippocampal CA1 area drives the oscillatory entrainment of PFC at neonatal age (*Brockmann et al., 2011*). Here, we developed a methodological approach to optically manipulate the neonatal HP along its septo-temporal axis. We provide causal evidence that theta frequency-specific activation of pyramidal neurons in the CA1 area of intermediate and ventral (i/

vHP), but not of dorsal HP (dHP) elicits broad band oscillations in the PFC of neonatal mice via dense axonal projections.

## Results

### Neonatal dorsal and intermediate/ventral hippocampus are differently entrained in discontinuous patterns of oscillatory activity

While different organization and function of dHP vs. i/vHP of adults have been extensively characterized (*Thompson et al., 2008*; *Dong et al., 2009*; *Patel et al., 2013*), their patterns of structural and functional maturation are still poorly understood. To fill this knowledge gap, we firstly examined the network oscillatory and firing activity of CA1 area of either dHP or i/vHP by performing extracellular recordings of the local field potential (LFP) and multiple unit activity (MUA) in neonatal [postnatal day (P) 8–10] non-anesthetized and urethane-anesthetized mice (n = 153). While urethane anesthesia led to an overall decrease of amplitude and power of oscillatory activity when compared to the non-anesthetized state of the same group of pups, the firing rate and timing as well as the synchrony and interactions within prefrontal-hippocampal networks were similar during both states (*Figure 1— figure supplement 1*). Due to the close proximity and the absence of reliable anatomical and functional borders between iHP and vHP at neonatal age, data from the two areas were pooled and referred as from i/vHP. The entire investigation was performed at the age of initiation of coupling between HP and PFC, that is, P8-10 (*Brockmann et al., 2011*). Independent of the position along the dorsal-ventral axis, the CA1 area was characterized by discontinuous oscillations with main frequency in theta band (4–12 Hz) and irregular low amplitude beta-gamma band components, which have been previously categorized as theta oscillations (*Brockmann et al., 2011*). They were accompanied by prominent sharp-waves (SPWs) reversing across the pyramidal layer (str. pyr.) and by strong MUA discharge (*Figure 1A and E*). While the general patterns of activity were similar in dHP and i/vHP, their properties significantly differed between the sub-divisions. The theta bursts in i/vHP had significantly higher occurrence (i/vHP: 8.1 ± 0.2 oscillations/min, n = 103 mice vs. dHP: 5.2 ± 0.3 oscillations/min, n = 41 mice; p<0.001), larger amplitude (i/vHP:110.6 ± 5.6 µV vs. dHP: 92.9 ± 2.6 µV; p=0.015), and shorter duration (i/vHP: 3.5 ± 0.1 s vs. dHP: 4.3 ± 0.1 s, p<0.001) when compared with dHP (*Figure 1B*, *Figure 1—figure supplement 2A*). Investigation of the spectral composition of theta bursts revealed significant differences within theta band with a stronger activation of dHP (relative power: dHP: 13.0 ± 1.3, n = 41 mice; i/vHP: 10.3 ± 0.5, n = 103 mice; p=0.026), whereas the faster frequency components were similar along the septo-temporal axis (relative power: 12–30 Hz: dHP, 15.0 ± 1.6, n = 41 mice; i/vHP, 13.2 ± 0.7 n = 103 mice, p=0.22; 30–100 Hz: dHP, 6.3 ± 0.6, n = 41 mice; i/vHP: 5.2 ± 0.3, n = 103 mice; p=0.073) (*Figure 1C*, *Figure 1—figure supplement 2B*).

Differences along the septo-temporal axis were detected both in hippocampal spiking and population events SPWs. Overall, pyramidal neurons in i/vHP fired at higher rates (0.45 ± 0.01 Hz, n = 557 units from 103 mice) than in the dHP (0.35 ± 0.02 Hz, n = 158 units from 41 mice; p=0.025) (*Figure 1D*). SPW in neonatal HP were more prominent in the dHP (712.8 ± 31.5 µV, n = 41 mice) when compared with those occurring in the i/vHP (223.8 ± 6.3 µV, n = 103 mice, p<0.001), yet their occurrence increased along the septo-temporal axis (dHP: 6.6 ± 0.5, n = 41 mice; i/vHP: 8.6 ± 0.2, n = 103 mice, p<0.001) (*Figure 1E and F*, *Figure 1—figure supplement 2D*. In line with our previous results (*Brockmann et al., 2011*), SPWs were accompanied by prominent firing centered around the SPW peak (dHP, 232 units; i/vHP, 670 units) that were phase-locked to hippocampal ripples (*Figure 1—figure supplement 2C*). The power of ripples decreased along the septo-temporal axis (relative power: dHP, 24.4 ± 3.3, n = 41 mice; i/vHP, 6.1 ± 0.60, n = 103 mice, p<0.001) (*Figure 1G,H*). Similarly, the ripple-related spiking was stronger in dHP when compared with i/vHP (peak firing: dHP: 1.13 ± 0.09 Hz, n = 232 units; i/vHP 0.84 ± 0.03 Hz, n = 670, p<0.001) (*Figure 1I and J*).

These data show that the activity patterns in the dorsal and intermediate/ventral CA1 area differ in their properties and spectral structure.

### Theta activity within dorsal and intermediate/ventral hippocampus differently entrains the neonatal prelimbic cortex

The different properties of network and neuronal activity in dHP vs. i/vHP led us to question their outcome for the long-range coupling in the developing brain. Past studies identified tight

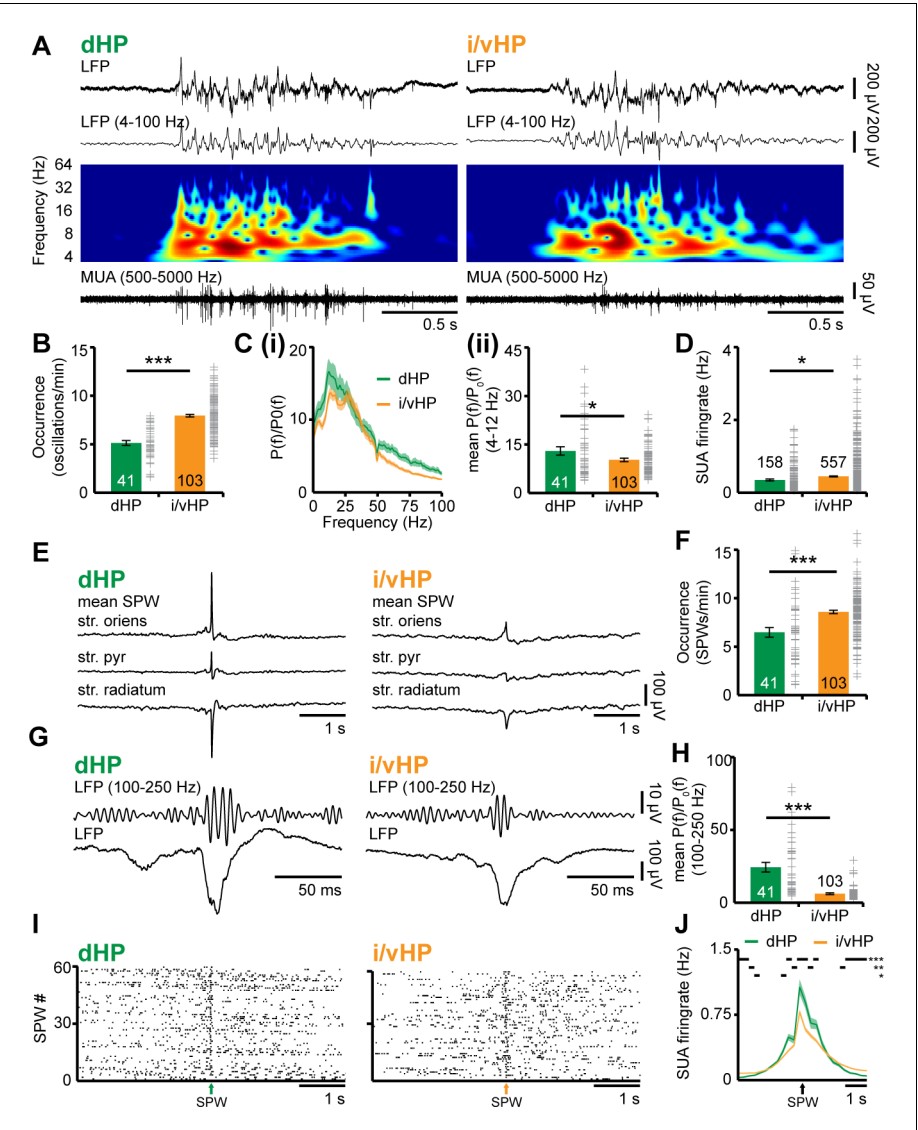

**Figure 1.** Patterns of discontinuous oscillatory activity in the CA1 area of the neonatal dHP and i/vHP in vivo. (A) Characteristic theta burst activity recorded in the CA1 area of the dHP (left) and i/vHP (right) of a P9 mouse displayed after band-pass filtering (4–100 Hz) and the corresponding MUA (500–5000 Hz). Color-coded frequency plots show the wavelet spectrum of LFP at identical time scale. (B) Bar diagram (mean ±SEM) displaying the occurrence of discontinuous theta bursts in dHP (n = 41 mice) and i/vHP (n = 103 mice). (C) Power analysis of discontinuous oscillatory activity P(f) normalized to the non-oscillatory period $P_0(f)$ in dHP and i/vHP. (i) Power spectra (4–100 Hz) averaged for all investigated mice. (ii) Bar diagrams quantifying the mean power within theta frequency band (4–12 Hz) in dHP (n = 41 mice) and i/vHP (n = 103 mice) (D) Bar diagram displaying the SUA of dHP (n = 158 units) and i/vHP (n = 557 units) after clustering of spike shapes. (E) Characteristic SPWs and ripple events recorded in dHP (left) and i/vHP (right). (F) Bar diagrams (mean ±SEM) displaying the SPWs occurrence in dHP and i/vHP. (G) Characteristic SPW-ripple events recorded in dHP (left) and i/vHP (right). (H) Bar diagram displaying the mean power of ripples in dHP and i/vHP. (I) Spike trains from neurons in dHP (left) and i/vHP (right) aligned to SPWs. (J) Histograms of SUA aligned to SPWs (n = 232 units for dHP, n = 670 for i/vHP). Data are represented as mean ± SEM. *p<0.05, **p<0.01, ***p<0.001.

DOI: https://doi.org/10.7554/eLife.33158.003

The following figure supplements are available for figure 1:

**Figure supplement 1.** Properties of network and neuronal activity in i/vHP of neonatal non-anesthetized and urethane-anesthetized mice.

DOI: https://doi.org/10.7554/eLife.33158.004

**Figure supplement 2.** Properties of network and neuronal activity in dHP vs. i/vHP of neonatal mice.

*Figure 1 continued on next page*

*Figure 1 continued*

DOI: https://doi.org/10.7554/eLife.33158.005

interactions between HP and PFC, which emerge already at neonatal age (*Brockmann et al., 2011*; *Hartung et al., 2016*) and are in support of memory at adulthood (*Krüger et al., 2012*; *Spellman et al., 2015*; *Place et al., 2016*). The discontinuous theta oscillations in HP have been proposed to drive the activation of local circuits in the PFC. To assess the coupling of dHP and i/vHP with PFC, we recorded simultaneously LFP and MUA in the corresponding hippocampal CA1 area and the prelimbic subdivision (PL) of the PFC of P8-10 mice. The entire investigation focused on PL, since in adults it is the prefrontal subdivision with the most dense innervation from HP (*Jay and Witter, 1991*; *Vertes et al., 2007*). In a first step, we examined the temporal correspondence of discontinuous oscillations recorded simultaneously in the PL and dHP, as well as in the PL and i/vHP. We previously characterized the network activity in the PL and showed that spindle-shaped oscillations switching between theta (4–12 Hz) and beta-gamma (12–40 Hz) frequency components alternate with periods of silence (*Brockmann et al., 2011*; *Cichon et al., 2014*; *Bitzenhofer et al., 2015*). The majority of prelimbic and hippocampal oscillations co-occurred within a narrow time window (*Figure 2A*). The temporal synchrony between prelimbic and hippocampal oscillations was assessed by performing spectral coherence analysis (*Figure 2B*). The results revealed a stronger coupling for PL-i/vHP (4–12 Hz: 0.17 ± 0.0069; 12–30 Hz: 0.31 ± 0.011; 30–100 Hz: 0.11 ± 0.0069, n = 103 mice) when compared with PL-dHP (4–12 Hz: 0.12 ± 0.0081; 12–30 Hz: 0.18 ± 0.0094; 30–100 Hz: 0.084 ± 0.004, n = 41 mice). In line with previous investigations, this level of coherence is a genuine feature of investigated neonatal networks and not the result of non-specific and conduction synchrony, since we considered only the imaginary component of the coherence spectrum, which excludes zero time-lag synchronization (*Nolte et al., 2004*).

Due to the symmetric interdependence of coherence, it does not offer reliable insights into the information flow between two brain areas. Therefore, in a second step, we estimated the strength of directed interactions between PL and HP by calculating the generalized partial directed coherence (gPDC) (*Baccala et al., 2007*; *Rodrigues and Baccala, 2016*) (*Figure 2C*). The method bases on the notion of Granger causality (*Granger, 1980*) and avoids distorted connectivity results due to different scaling of data in HP and PL (*Baccala et al., 2007*; *Taxidis et al., 2010*). Independent of the position along the septo-temporal axis, the information flow in theta or beta frequency band from either dorsal or intermediate/ventral HP to PL was significantly stronger than in the opposite direction. However, mean gPDC values for i/vHP → PL were significantly (p<0.001) higher (0.069 ± 0.003, n = 103 mice) when compared with those for dHP → PL (0.053 ± 0.003, n = 41 mice). Cross-correlation analysis confirmed these results (*Figure 2—figure supplement 1*). The stronger information flow from i/vHP to PL was confined to theta frequency range and was not detected for 12–30 Hz frequencies (i/vHP → PL: 0.048 ± 0.001; dHP → PL: 0.043 ± 0.002, p=0.16). Correspondingly, the firing of individual prelimbic neurons was precisely timed by the phase of oscillations in i/vHP but not dHP (*Figure 2D*). Almost 20% of clustered units (52 out of 310 units) were locked to theta phase in i/vHP, whereas only 6.5% of units (3 out of 46 units) were timed by dHP. The low number of locked cells in dHP precluded the comparison of coupling strength between the two hippocampal sub-divisions.

These results indicate that the distinct activity patterns in dHP and i/vHP at neonatal age have different outcomes in their coupling with the PL. Despite higher power, theta oscillations in dHP do not substantially account for prelimbic activity. In contrast, i/vHP seems to drive neuronal firing and network entrainment in the PL.

## SPWs-mediated output of intermediate/ventral but not dorsal hippocampus times network oscillations and spiking response in the neonatal prelimbic cortex

Since SPWs and ripples in dHP significantly differ from those in i/vHP, they might have a distinct impact on the developing PFC. While abundant literature documented the contribution of SPWs-spindles complex to memory-relevant processing in downstream targets, such as PFC (*Colgin, 2011*; *Buzsáki, 2015*; *Colgin, 2016*), it is unknown how these complexes affect the development of cortical activation. Simultaneous recordings from neonatal CA1 area either in dHP or i/vHP and PL showed

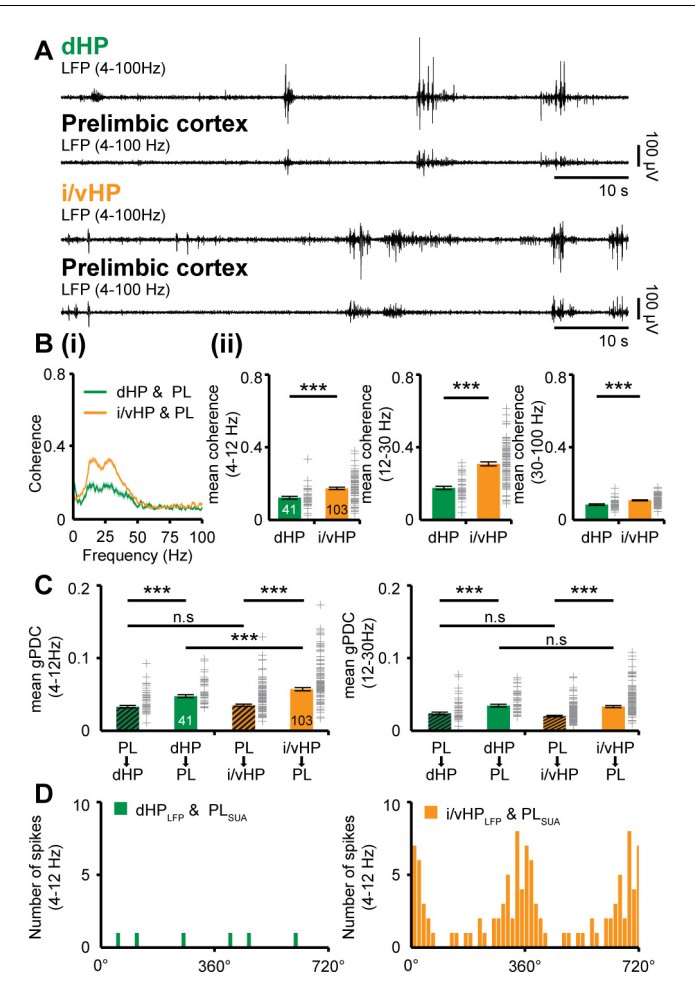

**Figure 2.** Dynamic coupling of hippocampal and prefrontal oscillatory activity along septo-temporal axis during neonatal development. (**A**) Simultaneous LFP recordings of discontinuous oscillatory activity in dHP and PL (top) and i/vHP and PL (bottom). (**B**) Long-range synchrony within prefrontal-hippocampal networks. (i) Average coherence spectra for simultaneously recorded oscillatory events in dHP and PL as well as i/vHP and PL. (ii) Bar diagrams (mean ±SEM) displaying the coherence in theta (4–12 Hz), beta (12–30 Hz), and gamma (30–100 Hz) band when averaged for all investigated mice. (**C**) Directed interactions between PL and either dHP or i/vHP monitored by general Partial Directed Coherence (gPDC). Bar diagrams displaying the gPDC calculated for theta (4–12 Hz, left) and beta (12–30 Hz, right) frequency and averaged for all investigated animals (n = 41 mice for dHP and PL, n = 103 mice for i/vHP and PL). (**D**) Histograms displaying the phase-locking of prelimbic spikes to theta oscillations in dHP (left) and i/vHP (right). Note the different proportion of spikes significantly locked along the septo-temporal axis (dHP, 3 of 46 units; i/vHP, 52 of 310 units). Data are represented as mean ± SEM. *p<0.05, ***p<0.001.

DOI: https://doi.org/10.7554/eLife.33158.006

The following figure supplement is available for figure 2:

**Figure supplement 1.** Cross-correlation of the amplitudes of band pass (4–12 Hz)-filtered LFP recorded from dHP and PL (green) as well as from i/vHP and PL (orange).

DOI: https://doi.org/10.7554/eLife.33158.007

that already at neonatal age, prefrontal oscillations are generated shortly (~100 ms) after hippocampal SPWs-ripples. This prelimbic activation is significantly stronger when induced by SPWs-ripples emerging in i/vHP than in dHP as reflected by the significantly higher power of oscillatory activity in theta (PL for dHP: 186.9 ± 12.5 $\mu V^2$; PL for i/vHP: 249.5 ± 14.5 $\mu V^2$, p=0.0088), beta (PL for dHP: 34.3 ± 3.3 $\mu V^2$; PL for i/vHP: 48.1 ± 2.8 $\mu V^2$, p=0.0049), and gamma (PL for dHP: 11.3 ± 0.9 $\mu V^2$; PL for i/vHP: 17.4 ± 1.2 $\mu V^2$, p=0.0026) frequency band (**Figure 3A**). The SPWs-ripple-induced

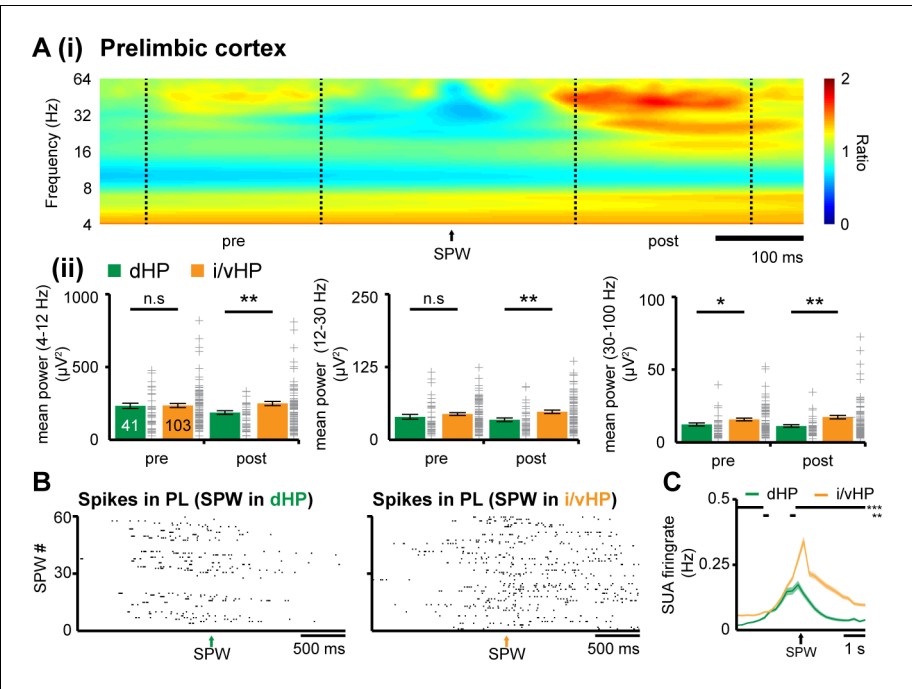

**Figure 3.** Coupling between neonatal PFC and HP during hippocampal SPWs. (**A**) Power changes in the PL during hippocampal SPWs. (i) Color-coded frequency plot showing the relative power in the PL aligned to the onset of SPWs detected in i/vHP when normalized to the power change caused in the PL by SPWs in the dHP. All other colors than green represent power augmentation (red) or decrease (blue). (ii) Bar diagrams displaying mean power changes of prelimbic activity in different frequency bands (left, theta; middle, beta; right, gamma) before (pre) and after (post) hippocampal SPWs in the dHP and i/vHP (n = 41 mice for dHP, n = 103 mice for i/vHP). (**B**) Spike trains recorded in the PL before and after SPWs occurring either in the dHP (left) or i/vHP (right). (**C**) Histograms of prelimbic spiking in relationship with hippocampal SPWs (n = 148 units for dHP, n = 560 units for i/vHP). Data are represented as mean ± SEM. *p<0.05, **p<0.01, ***p<0.001.
DOI: https://doi.org/10.7554/eLife.33158.008

The following figure supplement is available for figure 3:

**Figure supplement 1.** Phase-locking of SUA in PL before (pre) and after (post) SPWs detected in dHP (top, green) and i/vHP (bottom, orange).
DOI: https://doi.org/10.7554/eLife.33158.009

oscillatory activity in the PL of neonatal mice was accompanied by augmentation of firing rates. While the induced firing in i/vHP peaked (≈90 ms) after SPWs-ripples and remained significantly (p<0.001) elevated for several seconds, a less prominent peak was observed following SPW-ripples in dHP (*Figure 3B and C*). The phase-locking of prelimbic units was similar before and after SPWs (*Figure 3—figure supplement 1*).

These data reveal that SPWs-ripples from intermediate/ventral but less from the dorsal part of hippocampal CA1 correlate with pronounced neuronal firing and local entrainment in the PL of neonatal mice.

## Pyramidal neurons in intermediate/ventral but not dorsal hippocampus densely project to the prefrontal cortex at neonatal age

To identify the anatomical substrate of different coupling strength between i/vHP - PL and dHP - PL, we monitored the projections that originate from the CA1 area in both hippocampal subdivisions and target the PFC. The direct unilateral projections from hippocampal CA1 area to PL have been extensively investigated in adult brain (*Swanson, 1981*; *Jay and Witter, 1991*; *Vertes et al., 2007*) and are present already at neonatal age (*Brockmann et al., 2011*; *Hartung et al., 2016*). We tested for sub-division-specific differences by using retrograde and anterograde tracing. First, we injected unilaterally small amounts of the retrograde tracer Fluorogold (FG) into the PL of P7 mice (n = 8

mice). Three days after FG injections, labeled cells were found in str. pyr. of CA1 in both dHP and i/vHP (*Figure 4A*). However, their density was significantly different (p<0.001); whereas in dHP very few cells were retrogradely labeled ($0.15*10^3 \pm 0.074*10^3$ cells/mm$^2$), a large proportion of pyramidal-shaped cells in the CA1 area of i/vHP projects to PL ($3.29*10^3 \pm 0.19*10^3$ cells/mm$^2$).

Second, the preferential innervation of PL by pyramidal neurons from CA1 area of i/vHP was confirmed by anterograde staining with BDA (n = 9 mice). Small amounts of BDA were injected into the CA1 area of i/vHP (*Figure 4B*). They led to labeling of the soma and arborized dendritic tree of pyramidal neurons in str. pyr. with the characteristic orientation of axons. In 7 out of 9 mice anterogradely-labeled axons were found in the PL, preferentially within its deep layers V and VI.

Thus, the dense axonal projections from CA1 area of i/vHP might represent the substrate of HP-induced oscillatory entrainment of prelimbic circuits.

## Selective light manipulation of pyramidal neurons and interneurons in CA1 area of intermediate/ventral but not dorsal hippocampus causes frequency-specific changes in the oscillatory entrainment of neonatal prelimbic circuits

The tight coupling by synchrony and the directed information flow from hippocampal CA1 area to PL via direct axonal projections suggest that the HP acts already at neonatal age as a drive for prelimbic activation. Moreover, the differences identified between the dHP – PL and i/vHP – PL communication argue for prominent augmentation of driving force along the septo-temporal hippocampal

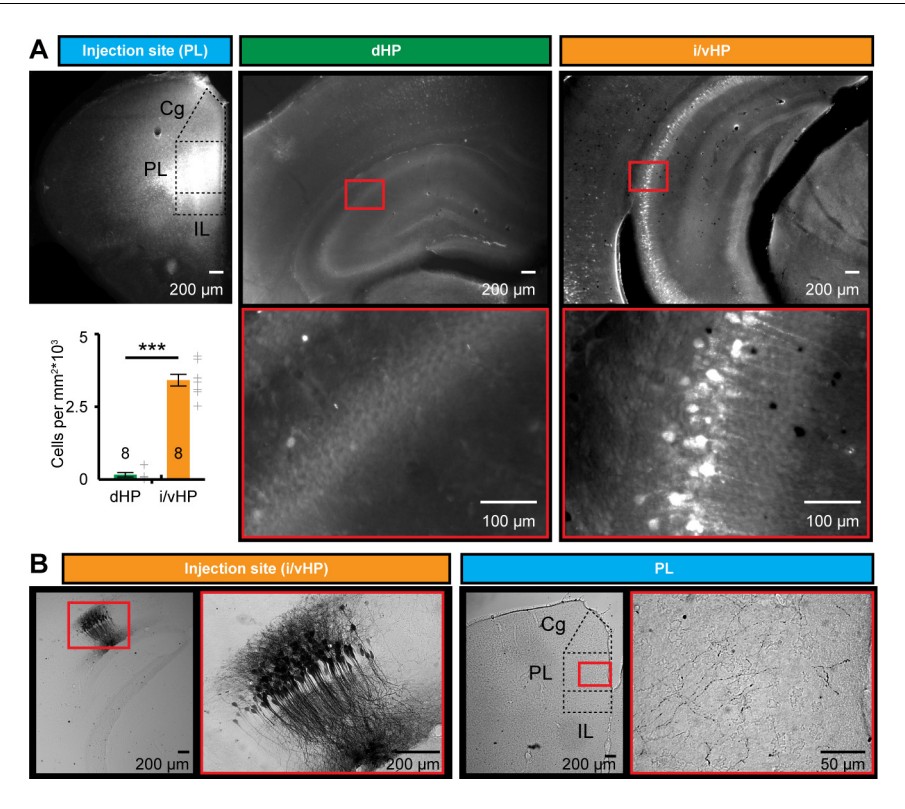

**Figure 4.** Long-range monosynaptic axonal projections connecting the neonatal PFC and hippocampal CA1 area along the septo-temporal axis. (**A**) Photomicrographs depicting dense retrogradely labelled neurons in the CA1 area of i/vHP (right) but not dHP (middle) after FG injection into PL at P1 (left). Bar diagram displays the overall density of retrogradely stained neurons when averaged for all investigated pups (n = 8 mice). (**B**) Photomicrographs depicting anterogradely labeled axons targeting the PL of a P10 mouse (right) after iontophoretic BDA injection into the CA1 area of i/vHP at P7 (left). The site of injection and the area with the highest axonal density are depicted at higher magnification. Data are represented as mean ± SEM. ***p<0.001.
DOI: https://doi.org/10.7554/eLife.33158.010

axis. To causally confirm these correlative evidences, we selectively activated by light the pyramidal neurons in the CA1 area of either dHP or i/vHP that had been transfected with a highly efficient fast-kinetics double mutant ChR2E123T/T159C (ET/TC) (*Berndt et al., 2011*) and the red fluorescent protein tDimer2 by *in utero* electroporation (IUE) (*Figure 5—figure supplement 1A*). This method enables stable area and cell type-specific transfection of neurons already prenatally without the need of cell-type specific promotors of a sufficiently small size (*Baumgart and Grebe, 2015*; *Szczurkowska et al., 2016*). To target neurons along the septo-temporal axis, distinct transfection protocols were used. When the IUE was performed with two paddles placed 25° leftward angle from the midline and a 0° angle downward from anterior to posterior, tDimer-positive neurons were mainly found in the CA1 area of the dHP, as revealed by the analysis of consecutive coronal sections from IUE-transfected P8-10 mice. Targeting of i/vHP succeeded only when three paddles were used, with both positive poles located at 90° leftward angle from the midline and the third negative pole at 0° angle downward from anterior to posterior (*Figure 5A*, S2B). Staining with NeuN showed that a substantial proportion of neurons in str. pyr. of CA1 area (dHP: $18.3 \pm 1.0\%$; n = 36 slices from 13 mice; i/vHP: $14.5 \pm 1.5\%$, n = 12 slices from 11 mice) were transfected by IUE. The shape of tDimer2-positive neurons, the orientation of primary dendrites, and the absence of positive staining for GABA confirmed that the light-sensitive protein ChR2(ET/TC) was integrated exclusively into cell lineages of pyramidal neurons (*Figure 5A*). Omission of ChR2(ET/TC) from the expression construct (i.e. opsin-free) yielded similar expression rates and distribution of tDimer2-positive neurons (*Figure 5—figure supplement 1C*).

To exclude non-specific effects of transfection procedure by IUE on the overall development of mice, we assessed the developmental milestones and reflexes of electroporated opsin-expressing and opsin-free mice (*Figure 5—figure supplement 1D*). While IUE caused significant reduction of litter size (non-electroporated $6.5 \pm 0.7$ pups/litter, electroporated: $4.5 \pm 0.5$ pups/litter, p=0.017), all investigated pups had similar body length, tail length, and weight during early postnatal period. Vibrissa placing, surface righting and cliff aversion reflexes were also not affected by IUE or transfection of neurons with opsins. These data indicate that the overall somatic development during embryonic and postnatal stage of ChR2(ET/TC)-transfected mice is unaltered.

We first assessed the efficiency of light stimulation in evoking action potentials in hippocampal pyramidal neurons in vivo. Blue light pulses (473 nm, 20–40 mW/mm$^2$) at different frequencies (4, 8, 16 Hz) led shortly (<10 ms) after the stimulus to precisely timed firing of transfected neurons in both dHP and i/vHP. Our previous experimental data and modeling work showed that the used light power did not cause local tissue heating that might interfere with neuronal spiking (*Stujenske et al., 2015*; *Bitzenhofer et al., 2017b*). For both hippocampal sub-divisions the efficiency of firing similarly decreased with augmenting frequency (*Figure 5B*). For stimulation frequencies >16 Hz, the firing lost the precise timing by light, most likely due to the immaturity of neurons and their projections.

To decide whether activation of HP boosts the entrainment of prelimbic circuits, we simultaneously performed multi-site recordings of LFP and MUA in PL and HP during pulsed light stimulation of CA1 area of dHP (n = 22 mice) or i/vHP (n = 9 mice) (*Figure 5C*). The firing in i/vHP timed by light at 8 Hz, but not at 4 Hz or 16 Hz, caused significant (theta: p=0.039, beta: p=0.030, gamma: p=0.0036) augmentation of oscillatory activity in all frequency bands as reflected by the higher power in the PL during the stimulation when compared with the time window before the train of pulses (*Figure 5D*, *Table 1*). In contrast, stimulation by light of dHP left the prelimbic activity unaffected. In opsin-free animals, stimulation of dHP and i/vHP led to no significant changes in the oscillatory activity (*Figure 5—figure supplement 2A*, *Table 1*). Rhythmic firing of prelimbic neurons was not detected after light activation of hippocampal subdivisions, most likely because hippocampal axons were rather sparse.

To confirm the driving role of i/vHP for the generation of oscillatory activity in PL, we selectively transfected Dlx5/6 positive interneurons with either ChETA or archaerhodopsin (ArchT). Blue light stimulation (473 nm) confined to i/vHP of Dlx5/6–ChETA mice (n = 19) led to a significant reduction of hippocampal power in all frequency bands (theta: p=0.024, beta: p=0.018, gamma: p=0.044). Correspondingly, the oscillatory activity in PL diminished (theta: p=0.027, beta: p=0.077, gamma: p=0.019) (*Figure 6A,B*). Silencing of interneurons in Dlx5/6-ArchT mice (n = 13) by yellow light (600 nm) had an opposite effect and caused augmentation of oscillatory activity both within i/vHP (theta:

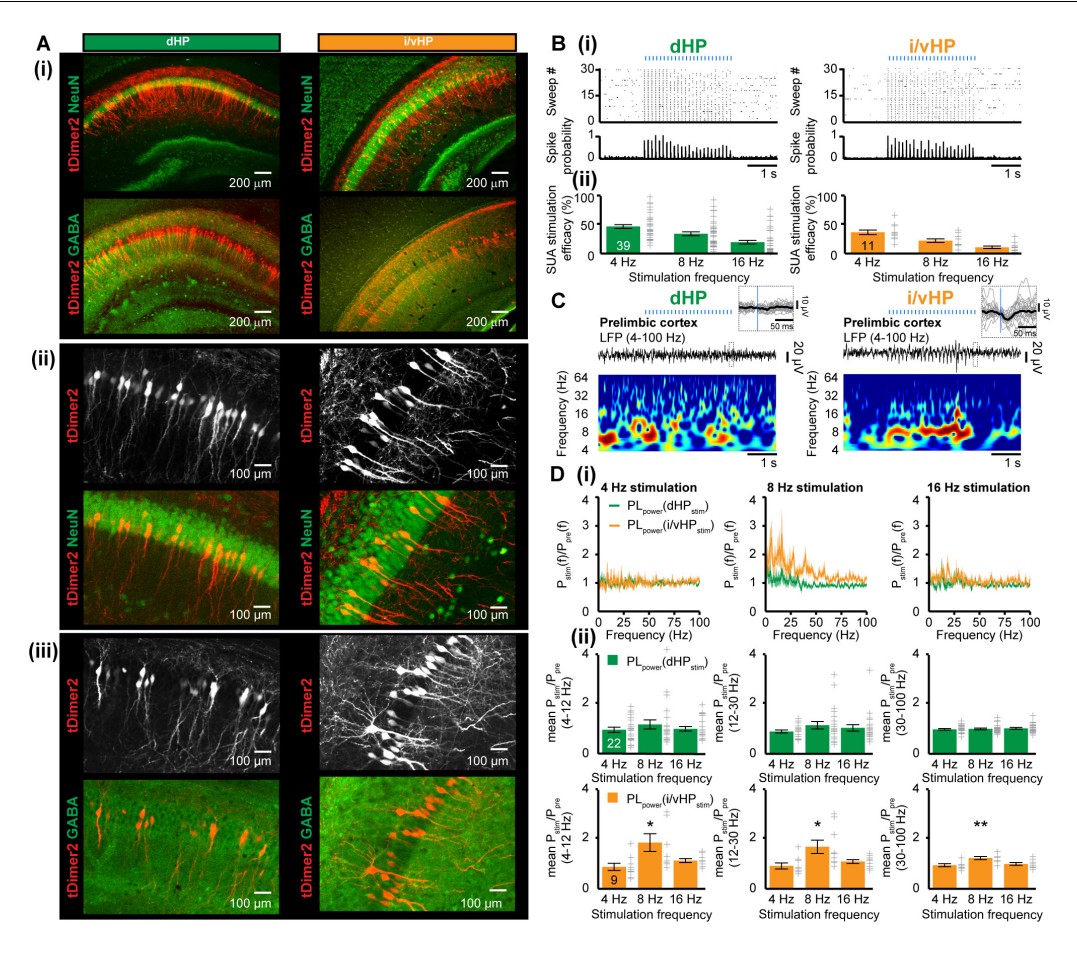

**Figure 5.** Optogenetic activation of pyramidal neurons in the CA1 area of dHP and i/vHP has different effects on the network activity of neonatal PL. (A) Cell- and layer-specific transfection of dHP or i/vHP with CAG-ChR2(ET/TC)–2A-tDimer2 by site-directed IUE. (i) Photomicrographs depicting tDimer2-expressing pyramidal neurons (red) in the CA1 region of dHP (left) and i/vHP (right) when stained for NeuN (green, top panels) or GABA (green, bottom panels). (ii) Photomicrographs depicting the transfected hippocampal neurons when co-stained for NeuN and displayed at larger magnification. (iii) Photomicrographs depicting transfected hippocampal neurons when co-stained for GABA and displayed at larger magnification. (B) Optogenetic activation of pyramidal neurons in CA1 area along septo-temporal axis. (i) Representative raster plot and corresponding spike probability histogram for dHP (left) and i/vHP (right) in response to 30 sweeps of 8 Hz pulse stimulation (3 ms pulse length, 473 nm). (ii) Bar diagram displaying the efficacy of inducing spiking in dHP and i/vHP of different stimulation frequencies. (C) Characteristic light-induced discontinuous oscillatory activity in the PL of a P10 mouse after transfection of pyramidal neurons in the CA1 area of the dHP (left) or i/vHP (right) with ChR2(ET/TC) by IUE. The LFP is displayed after band-pass filtering (4–100 Hz) together with the corresponding color-coded wavelet spectrum at identical time scale. Inset, individual (gray) and averaged (black) prelimbic LFP traces displayed at larger time scale in response to light stimulation in HP. (D) Power analysis of prelimbic oscillatory activity $P_{stim}(f)$ after light stimulation of dHP (green) and i/v HP (orange) at different frequencies (4, 8, 16 Hz) normalized to the activity before stimulus $P_{pre}(f)$. (i) Power spectra (0–100 Hz) averaged for all investigated mice. (ii) Bar diagrams displaying mean power changes in PL during stimulation of either dHP (top panels) or i/vHP (bottom panels). Data are represented as mean ± SEM. *p<0.05, **p<0.01.

DOI: https://doi.org/10.7554/eLife.33158.011

The following figure supplements are available for figure 5:

**Figure supplement 1.** Experimental protocol for *in utero* electroporation of the hippocampus.
DOI: https://doi.org/10.7554/eLife.33158.012

**Figure supplement 2.** Response in prelimbic cortex for opsin-free animals.
DOI: https://doi.org/10.7554/eLife.33158.013

p<0.001, beta: p=0.0012, gamma: p<0.001) and PL (theta: p<0.001, beta: p<0.001, gamma: p<0.001) (*Figure 6A,C*).

**Table 1.** Mean power changes in PL after light stimulation of dHP or i/vHP in ChR2(ET/TC)-containing and opsin-free animals. *p<0.05, **p<0.01.

| | dHP | | | I/vHP | | |
|---|---|---|---|---|---|---|
| | **Stimulation frequency** | | | **Stimulation frequency** | | |
| **ChR2(ET/TC)** | 4 Hz | 8 Hz | 16 Hz | 4 Hz | 8 Hz | 16 hz |
| Theta | 0.97 ± 0.10 | 1.19 ± 0.19 | 1.0 ± 0.093 | 0.90 ± 0.15 | 1.89 ± 0.36 (*) | 1.16 ± 0.08 |
| Beta | 0.91 ± 0.06 | 1.17 ± 0.15 | 1.06 ± 0.13 | 0.94 ± 0.12 | 1.72 ± 0.27 (*) | 1.12 ± 0.08 |
| Gamma | 1.0 ± 0.035 | 1.00 ± 0.19 | 1.04 ± 0.38 | 0.97 ± 0.06 | 1.26 ± 0.06 (**) | 1.02 ± 0.06 |
| **Opsinfree** | | | | | | |
| Theta | 1.11 ± 0.14 | 1.09 ± 0.19 | 1.14 ± 0.22 | 1.17 ± 0.27 | 1.17 ± 0.20 | 1.16 ± 0.12 |
| Beta | 1.13 ± 0.15 | 0.99 ± 0.16 | 1.11 ± 0.11 | 1.05 ± 0.22 | 0.95 ± 0.18 | 1.08 ± 0.13 |
| Gamma | 1.08 ± 0.06 | 0.93 ± 0.04 | 1.03 ± 0.03 | 0.89 ± 0.09 | 0.94 ± 0.07 | 0.97 ± 0.04 |

DOI: https://doi.org/10.7554/eLife.33158.014

Taken together, these data reveal the critical role of hippocampal activity for the oscillatory entrainment of PL and identify pyramidal neurons in CA1 area of of i/vHP but not dHP as drivers for the broad activation of local prelimbic circuits.

## Discussion

Combining selective optogenetic activation with extracellular recordings and tracing of projections in neonatal mice in vivo, we provide causal evidence that theta activity in the CA1 area of i/vHP but not dHP drives network oscillations within developing prefrontal cortex. Despite stronger theta power in the dHP, solely optical activation of the pyramidal neurons in i/vHP at theta frequency range (8 Hz) boosted the emergence of discontinuous oscillatory activity in theta and beta-gamma bands in the neonatal PFC. These data identify the cellular substrate of the directed interactions between neonatal hippocampus and prefrontal cortex and offer new perspectives for the interrogation of long-range coupling in the developing brain and its behavioral readout.

### Distinct patterns of functional maturation in dorsal and intermediate/ventral hippocampus

The abundant literature dedicated to the adult hippocampus mainly deals with a single cortical module (*Amaral et al., 2007*). However, an increasing number of studies in recent years revealed distinct organization, processing mechanisms and behavioral relevance for dHP vs. i/vHP (*Fanselow and Dong, 2010*; *Bannerman et al., 2014*; *Strange et al., 2014*). For example, the dHP, which receives dense projections from the entorhinal cortex (*Witter and Amaral, 2004*), is mainly involved in spatial navigation (*O'Keefe and Nadel, 1978*; *Moser et al., 1995*; *Moser et al., 1998*). In contrast, the ventral part receives strong cholinergic and dopaminergic innervation (*Witter et al., 1989*; *Pitkänen et al., 2000*) and contributes to processing of non-spatial information (*Bannerman et al., 2003*; *Bast et al., 2009*). Correspondingly, the network and neuronal activity changes along the septo-temporal axis. The power of the most prominent activity pattern in the adult HP, the theta oscillations, as well as the theta timing of the neuronal firing was found to be substantially reduced in the i/vHP when compared with dHP (*Royer et al., 2010*). By these means, the precise spatial representation deteriorates along the septo-temporal axis, since theta activity is directly linked to place cell representation (*O'Keefe and Recce, 1993*; *Geisler et al., 2007*). In contrast, SPWs are more frequent and ripples have higher amplitude and frequency in the ventral HP than in the dHP (*Patel et al., 2013*).

Our data uncovered that some of these differences in the activity patterns along the septo-temporal axis emerge already during early neonatal development. Similar to findings from adult rodents, the power of theta bursts at neonatal age was higher in dHP than in i/vHP. The amplitude of SPWs and the power of ripples decreased along the septo-temporal axis. These findings give insights into the mechanisms underlying the early generation of activity patterns. It has been proposed that the differences in theta dynamics along the septo-temporal axis result from distinct innervation, on the

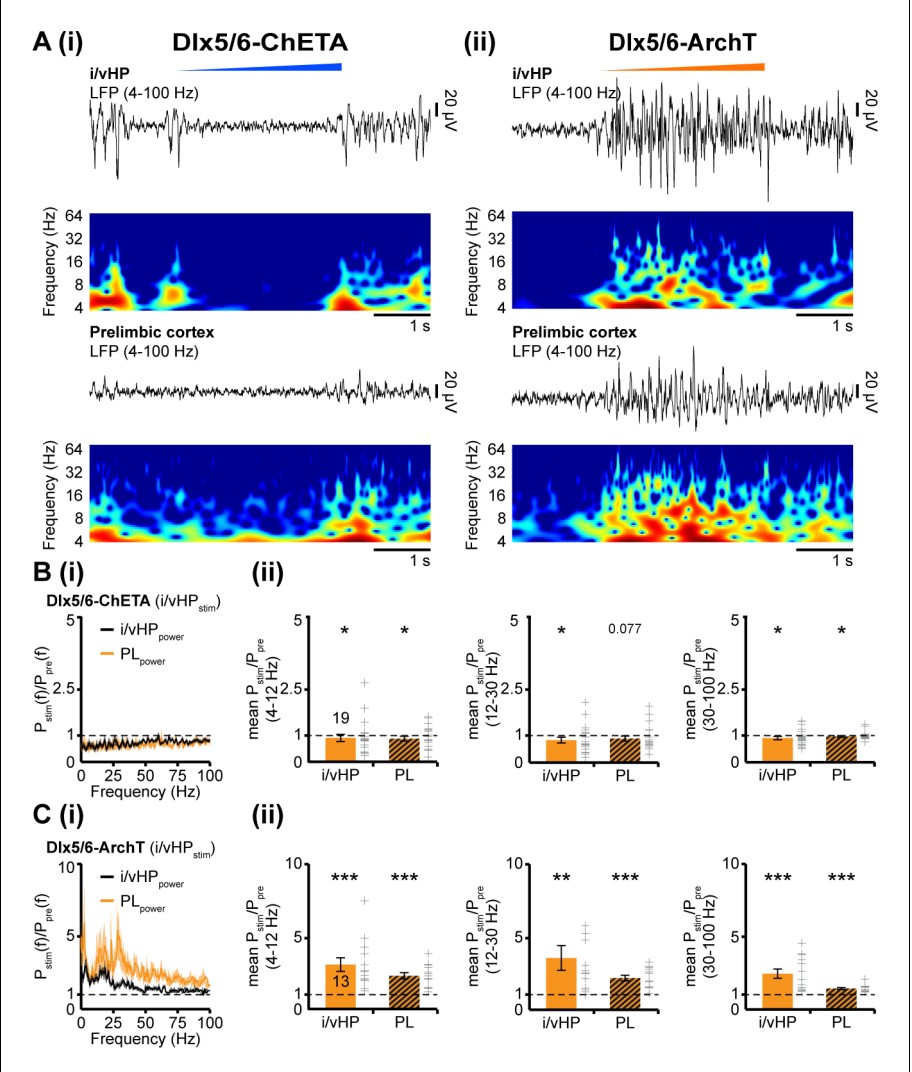

**Figure 6.** Modulation of oscillatory activity in i/vHP by optogenetic manipulation of interneurons affects the entrainment of neonatal PL. (**A**) Light-induced modulation of oscillatory activity in i/vHP and PL of a P9 mouse after transfection of interneurons in the CA1 area of the i/vHP with ChETA (i) or ArchT (ii). The LFP is displayed after band-pass filtering (4–100 Hz) together with the corresponding color-coded wavelet spectrum at identical time scale. (**B**) Power of oscillatory activity in i/vHP and PL after optogenetic activation of interneurons in i/vHP ($P_{stim}(f)$) normalized to the activity before stimulus $P_{pre}(f)$. (i) Power spectra (0–100 Hz) averaged for all investigated mice. (ii) Bar diagrams displaying mean power changes in theta, beta, and gamma frequency bands for the oscillations recorded in i/vHP and PL during light stimulation in i/vHP. (**C**) Same as (**B**) for silencing of ArchT-transfected interneurons in i/vHP by yellow light. Data are shown as mean ± SEM. *$p<0.05$, **$p<0.01$, ***$p<0.001$.
DOI: https://doi.org/10.7554/eLife.33158.015

one hand, and from specific intrinsic properties of hippocampal neurons, on the other hand. Cholinergic projections of different origin in the dHP and i/vHP (*Stewart and Fox, 1990*; *Amaral et al., 2007*) as well as maturational differences in the intrinsic resonant properties of hippocampal neurons and notable gradients of parvalbumin immunoreactivity along the septo-temporal axis (*Honeycutt et al., 2016*) may contribute to the observed differences.

Quantification along the septo-temporal axis revealed that, similar to adults, the occurrence of SPWs was higher in the i/vHP and their amplitude was larger in the neonatal dHP (*Patel et al., 2013*). It is still an issue of debate when exactly ripples emerge in the developing hippocampus, although it is obvious that they appear later than theta bursts and SPWs, most likely towards the end of the first and during second postnatal week (*Buhl and Buzsáki, 2005*; *Brockmann et al.,*

2011). Their underlying mechanisms at neonatal age remain also largely unknown and need to be related to age-dependent changes in gap junctional coupling and GABA switch (*Ben-Ari et al., 1989*; *Zhang et al., 1990*; *Yuste et al., 1995*). The organization of SPWs and ripples is of particular relevance when considering their impact on the early activity of PFC. Already at neonatal age, the prelimbic firing and oscillatory entrainment is timed by SPWs-ripples. Of note, the degree of timing varies along the septo-temporal axis and is much higher for the i/vHP.

## Optogenetic interrogation of long-range coupling in the developing brain

At adult age the communication between PFC and HP has been investigated in relationship with memory tasks both under physiological and disease-related conditions (*Sirota et al., 2008*; *Adhikari et al., 2010*; *Sigurdsson et al., 2010*; *Eichenbaum, 2017*). Depending on the phase of memory processing, the prefrontal-hippocampal coupling via oscillatory synchrony has been found to be either unidirectional from the HP to PFC or bidirectional (*Siapas et al., 2005*; *Hallock et al., 2016*; *Place et al., 2016*). Both theta and gamma network oscillations contribute to the functional long-range coupling. The model of prefrontal-hippocampal communication has been initially built based on experimental evidence correlating the temporal organization of neuronal and network activity in the two brain areas. The time delay between spike trains and oscillatory phase or between oscillations enabled to propose that the information flows in one direction or the other via mono- or polysynaptic axonal projections. More recently, a direct causal assessment of the coupling became possible through optogenetic interrogation of neural circuits. In a seminal study, Spellman and colleagues used light-driven inhibition of axonal terminals for dissecting the directionality of interactions between PFC and HP during different phases of memory retrieval (*Spellman et al., 2015*).

We previously showed that at neonatal age, long before full maturation of memory and attentional abilities, discontinuous theta bursts in i/vHP are temporally correlated to the network oscillations in the PFC and time the prefrontal firing (*Brockmann et al., 2011*; *Hartung et al., 2016*). Moreover, the temporal delay of 10–20 ms between prefrontal and hippocampal spike trains as well as the estimated directionality of information flow between the two areas suggested that hippocampal theta drives the oscillatory entrainment of the local circuits in the PFC. The present data directly prove this hypothesis, taking advantage of the recently developed protocol for optogenetic manipulation of neuronal networks at neonatal age (*Bitzenhofer et al., 2017a*, *2017b*). We observed that prelimbic circuits were effectively entrained when the stimulation of paramidal neurons in i/vHP occurred at 8 Hz but not at 4 Hz or 16 Hz. Such frequency-specific effect might result from intrinsic resonance properties of neurons mediated through hyperpolarization-activated cyclic nucleotide-gated (HCN) channels (*Hu et al., 2002*; *Stark et al., 2013*). It has been previously proposed that oscillatory activity in cortical regions may be entrained due to the rhythmic theta-band output from the hippocampus (*Stark et al., 2013*; *Colgin, 2016*).

Several considerations regarding the technical challenges of optogenetic manipulation of HP along the septo-temporal axis need to be made. Besides the inherent difficulties related to the specificity of promoters for selective transfection and the targeting procedure that are ubiquitary for all developing networks and have been addressed elsewhere (*Bitzenhofer et al., 2017a*), confinement of light-sensitive proteins to pyramidal neurons of either dHP or i/vHP required special attention. In a previous study (*Bitzenhofer et al., 2017b*), we developed a selective targeting protocol of neonatal neurons that relies on the combination of CAG promoter and IUE. By these means, the expression of light-sensitive proteins in the neurons located in the neocortical layer and area of interest was sufficiently high to ensure their reliable activation. Similarly, the expression of ChR2(ET/TC) in the pyramidal neurons of hippocampal CA1 area under the CAG promoter was sufficiently high to reliably cause network and neuronal activity. Taking into account that viral transduction, which usually requires 10–14 days for stable expression, is only of limited usability to investigate local network interactions during development, IUE seems to represent the method of choice for manipulating circuits at this early age. IUE enables targeting of precursor cells of neuronal and glial subpopulations, based on their distinct spatial and temporal patterns of generation in the ventricular zone (*Tabata and Nakajima, 2001*; *Borrell et al., 2005*; *Niwa et al., 2010*; *Hoerder-Suabedissen and Molnár, 2015*). IUE based on two electrode paddles enabled selective targeting of pyramidal neurons in the CA1 area of dHP in more than half of the pups per litter (*Figure 5—figure supplement 1*), but it completely failed (0 out of 32 mice) to target these neurons in i/vHP. Therefore, it was

necessary to use a modified IUE protocol based on three electrodes. This protocol, although more complicated and time consuming, allows reliable transfection at brain locations that are only able to be sporadically targeted by two electrodes The IUE-induced expression of light sensitive proteins enables the reliable firing of neurons in both dHP and i/vHP in response to light pulses. One intriguing question is how many pyramidal neurons in str. pyr. of CA1 area must be synchronously activated to drive the oscillatory entrainment of prelimbic circuitry. Anterograde and retrograde tracing demonstrated the density increase along the septo-temporal axis of hippocampal axons targeting the PL. Light activation/inhibition of these axonal terminals paired with monitoring of network oscillations in the PFC might offer valuable insights into the patterns of coupling sufficient for activation.

## Functional relevance of frequency-specific drive within developing prefrontal-hippocampal networks

Abundant literature links theta frequency coupling within prefrontal-hippocampal networks to cognitive performance and emotional states of adults (*Adhikari et al., 2010*; *Xu and Südhof, 2013*; *Spellman et al., 2015*; *Hallock et al., 2016*; *Place et al., 2016*; *Ye et al., 2017*). The early emergence of directed communication between PFC and i/vHP raises the question of functional relevance of this early coupling during development and at adulthood.

The maturation of cognitive abilities is a process even more protracted than sensory development and starts during second-third postnatal week (*Hanganu-Opatz, 2010*; *Cirelli and Tononi, 2015*). Some of these abilities, such as recognition memory, can be easily monitored at early age and seems to critically rely on structurally and functionally intact prefrontal-hippocampal networks (*Krüger et al., 2012*). Direct assessment of the role of neonatal communication for memory performance as performed for adult circuits is impossible due to the temporal delay of the two processes. The alternative is to manipulate the activity of either PFC, HP or the connectivity between them during defined developmental time windows and monitor the juvenile and adult consequences at structural, functional and behavioral levels. The present data and optogenetic protocol represent the prerequisite of this investigation, opening new perspectives for assessing the adult behavioral readout of long-range communication in the developing brain.

One question that remains to be addressed is how the hippocampal theta drive shapes the maturation of prefrontal-hippocampal networks. Following the general rules of activity-dependent plasticity (*Hubel et al., 1977*; *Huberman et al., 2006*; *Xu et al., 2011*; *Yasuda et al., 2011*), the precisely timed excitatory inputs from the i/vHP to the PL might facilitate the wiring of local prefrontal circuitry and enable the refinement of behaviorally relevant communication scaffold between the two regions. By these means, the prefrontal activity driven by projection neurons in the HP act as a template, having a pre-adaptive function that facilitates the tuning of circuits with regard to future conditions. This instructive role of theta activity for the prefrontal circuits needs to be proven by manipulation of temporal structure of the hippocampal drive without affecting the overall level of activity. Understanding the rules that govern the early organization of large-scale networks represents the pre-requisite for identifying the structural and functional deficits related to abnormal behavior and disease.

## Materials and methods

**Key resources table**

| Reagent type (species) or resource | Designation | Source or reference | Identifiers | Additional information |
|---|---|---|---|---|
| antibody | mouse monoclonal Alexa Fluor-488 conjugated antibody against NeuN | Merck Millipore | MAB377X | 1:200 dilution |
| antibody | rabbit polyclonal primary antibody against GABA | Sigma-Aldrich | A2052 | 1:1000 dilution |
| antibody | Alexa Fluor-488 goat anti-rabbit IgG secondary antibody | Merck Millipore | A11008 | 1:500 dilution |

*Continued on next page*

*Continued*

| Reagent type (species) or resource | Designation | Source or reference | Identifiers | Additional information |
|---|---|---|---|---|
| chemical compound, drug | Isoflurane | Abbott | B506 | |
| chemical compound, drug | Urethane | Fluka analytical | 94300 | |
| chemical compound, drug | Fluorogold | Fluorochome, LLC | 52–9400 | |
| chemical compound, drug | Biotinylated dextran amine, 10.000 MW | Thermo Fisher Scientific | D1956 | |
| commercial assay or kit | NucleoBond PC 100 | Macherey-Nagel | 740573 | |
| strain, strain background (mouse, both genders) | C57Bl/6J | Universitätsklinikum Hamburg-Eppendorf – Animal facility | C57Bl/6J | https://www.jax.org/strain/008199 |
| strain, strain background (mouse, both genders) | Tg(dlx5a-cre)1Mekk/J | The Jackson Laboratory | Tg(dlx5a-cre)1Mekk/J | https://www.jax.org/strain/017455 |
| strain, strain background (mouse, both genders) | R26-CAG-LSL-2XChETA-tdTomato | The Jackson Laboratory | R26-CAG-LSL-2XChETA-tdTomato | https://www.jax.org/strain/021188 |
| strain, strain background (mouse, both genders) | Ai40(RCL-ArchT/EGFP)-D | The Jackson Laboratory | Ai40(RCL-ArchT/EGFP)-D | |
| recombinant DNA reagent | pAAV-CAG-ChR2 (E123T/T159C)−2AtDimer2 | Provided by T. G. Oertner | pAAV-CAG-ChR2 (E123T/T159C)−2AtDimer2 | http://www.oertner.com/ |
| recombinant DNA reagent | pAAV-CAG-tDimer2 | Provided by T. G. Oertner | pAAV-CAG-tDimer2 | http://www.oertner.com/ |
| software, algorithm | Matlab R2015a | MathWorks | Matlab R2015a | https://www.mathworks.com |
| software, algorithm | Offline Sorter | Plexon | Offline Sorter | http://www.plexon.com/ |
| software, algorithm | ImageJ 1.48 c | ImageJ | ImageJ 1.48 c | https://imagej.nih.gov/ij/ |
| software, algorithm | SPSS Statistics 21 | IBM | SPSS Statistics 21 | https://www.ibm.com/analytics/us/en/technology/spss/ |
| software, algorithm | Cheetah 6 | Neuralynx | Cheetah 6 | http://neuralynx.com/ |
| other | Arduino Uno SMD | Arduino | A000073 | A000073 |
| other | Digital Lynx 4SX | Neuralynx | Digital Lynx 4SX | http://neuralynx.com/ |
| other | Diode laser (473 nm) | Omicron | LuxX 473–100 | |
| other | Electroporation device | BEX | CUY21EX | |
| other | Electroporation tweezer-type paddles | Protech | CUY650-P5 | |
| other | Recording electrode (1 shank, 16 channels) | Neuronexus | A1 × 16–3 mm-703-A16 | |
| other | Recording optrode (1 shank, 16 channels) | Neuronexus | A1 × 16–5 mm-703-OA16LP | |
| other | Digital midgard precision current source | Stoelting | 51595 | |

## Experimental model and subject details

### Mice

All experiments were performed in compliance with the German laws and the guidelines of the European Union for the use of animals in research and were approved by the local ethical committee (111/12, 132/12). Timed-pregnant C57Bl/6J mice from the animal facility of the University Medical Center Hamburg-Eppendorf were housed individually in breeding cages at a 12 hr light/12 hr dark cycle and fed *ad libitum*. The day of vaginal plug detection was defined E0.5, while the day of birth was assigned as P0. Both female and male mice underwent light stimulation and multi-site electrophysiological recordings at P8-10 after transfection with light-sensitive proteins by IUE at E15.5. For monitoring of projections, tracers were injected at P7 and monitored in their distribution along the axonal tracts at P10. For specifically addressing interneurons by light, the Dlx5/6-Cre drive line (Tg (dlx5a-cre)1Mekk/J, Jackson Laboratory) was crossed with either ArchT (Ai40(RCL-ArchT/EGFP)-D,

Jackson Laboratory) or ChETA (R26-CAG-LSL-2XChETA-tdTomato, Jackson Laboratory) reporter line.

## Methods details

### Surgical procedures

#### In utero electroporation

Starting one day before and until two days after surgery, timed-pregnant C57Bl/6J mice received on a daily basis additional wet food supplemented with 2–4 drops Metacam (0.5 mg/ml, Boehringer-Ingelheim, Germany). At E15.5 randomly assigned pregnant mice were injected subcutaneously with buprenorphine (0.05 mg/kg body weight) 30 min before surgery. The surgery was performed on a heating blanket and toe pinch and breathing were monitored throughout. Under isoflurane anesthesia (induction: 5%, maintenance: 3.5%) the eyes of the dam were covered with eye ointment to prevent damage before the uterine horns were exposed and moistened with warm sterile phosphate buffered saline (PBS, 37°C). Solution containing 1.25 µg/µl DNA [pAAV-CAG-ChR2(E123T/T159C)−2A-tDimer2, or pAAV-CAG-tDimer2)] (*Figure 5—figure supplement 1A*) and 0.1% fast green dye at a volume of 0.75–1.25 µl were injected into the right lateral ventricle of individual embryos using pulled borosilicate glass capillaries with a sharp and long tip. Plasmid DNA was purified with Nucleo-Bond (Macherey-Nagel, Germany). 2A encodes for a ribosomal skip sentence, splitting the fluorescent protein tDimer2 from the opsin during gene translation. Two different IUE protocols were used to target pyramidal neurons in CA1 area of either dHP or i/vHP. To target dHP, each embryo within the uterus was placed between the electroporation tweezer-type paddles (5 mm diameter, Protech, TX, USA) that were oriented at a 25° leftward angle from the midline and a 0° angle downward from anterior to posterior. Electrode pulses (35 V, 50 ms) were applied five times at intervals of 950 ms controlled by an electroporator (CU21EX, BEX, Japan) (*Figure 5—figure supplement 1B*(i)) (*Baumgart and Grebe, 2015*). To target i/vHP, a tri-polar approach was used (*Szczurkowska et al., 2016*). Each embryo within the uterus was placed between the electroporation tweezer-type paddles (5 mm diameter, both positive poles, Protech, TX, USA) that were oriented at 90° leftward angle from the midline and a 0° angle downward from anterior to posterior. A third custom build negative pole was positioned on top of the head roughly between the eyes. Electrode pulses (30 V, 50 ms) were applied six times at intervals of 950 ms controlled by an electroporator (CU21EX, BEX, Japan). By these means, neural precursor cells from the subventricular zone, which radially migrate into the HP, were transfected. Uterine horns were placed back into the abdominal cavity after electroporation. The abdominal cavity was filled with warm sterile PBS (37°C) and abdominal muscles and skin were sutured individually with absorbable and non-absorbable suture thread, respectively. After recovery, pregnant mice were returned to their home cages, which were half placed on a heating blanket for two days after surgery.

#### Retrograde and anterograde tracing

For retrograde tracing, mice were injected at P7 with Fluorogold (Fluorochrome, LLC, USA) unilaterally into the PFC using iontophoresis. The pups were placed in a stereotactic apparatus and kept under anesthesia with isoflurane (induction: 5%, maintenance: 2.5%) for the entire procedure. A 10 mm incision of the skin on the head was performed with small scissors. The bone above the PFC (0.5 mm anterior to bregma, 0.3 mm right to the midline) was carefully removed using a syringe. A glass capillary (≈20 µm tip diameter) was filled with ≈1 µL of 5% Fluorogold diluted in sterile water by capillary forces, and a silver wire was inserted such that it was in contact with the Fluorogold solution. For anterograde tracing, mice were injected at P7 with the anterograde tracer biotinylated dextran amine (BDA) (Thermo Fisher Scientific, USA) unilaterally into i/vHP using iontophoresis and surgery protocols as described above. The bone above i/vHP (0.7 mm anterior to lambda, 2.3 mm right to the midline) was carefully removed using a syringe. A glass capillary (≈30 µm tip diameter) was filled with ≈1 µL of 5% BDA diluted in 0.125 M phosphate buffer by capillary forces, and a silver wire was inserted such that it was in contact with the BDA solution. For both anterograde and retrograde tracing, the positive pole of the iontophoresis device was attached to the silver wire, the negative one was attached to the skin of the neck. The capillary was carefully lowered into the PFC (≈1.5 mm dorsal from the dura) or HP (≈1.5 mm dorsal from the dura). Iontophoretically injection by applying anodal current to the pipette (6 s on/off current pulses of 6 µA) was done for 5 min.

Following injection, the pipette was left in place for at least 5 min and then slowly retracted. The scalp was closed by application of tissue adhesive glue and the pups were left on a heating pad for 10–15 min to fully recover before they were given back to the mother. The pups were perfused at P10.

### Surgical preparation for acute electrophysiological recording and light delivery

For recordings in non-anesthetized state, 0.5% bupivacain/1% lidocaine was locally applied on the neck muscles. For recordings under anesthesia, mice were injected i.p. with urethane (1 mg/g body weight; Sigma-Aldrich, MO, USA) prior to surgery. For both groups, under isoflurane anesthesia (induction: 5%, maintenance: 2.5%) the head of the pup was fixed into a stereotaxic apparatus using two plastic bars mounted on the nasal and occipital bones with dental cement. The bone above the PFC (0.5 mm anterior to bregma, 0.5 mm right to the midline for layer V/VI), hippocampus (2.0 mm posterior to bregma, 1.0 mm right to the midline for dHP, 3.5 mm posterior to bregma, 3.5 mm right to the midline for i/vHP) was carefully removed by drilling a hole of <0.5 mm in diameter. After a 10–20 min recovery period on a heating blanket mice were moved to the setup for electrophysiological recording. Throughout the surgery and recording session the mouse was positioned on a heating pad with the temperature kept at 37°C.

### Perfusion

Mice were anesthetized with 10% ketamine (aniMedica, Germany)/2% xylazine (WDT, Germany) in 0.9% NaCl solution (10 µg/g body weight, i.p.) and transcardially perfused with Histofix (Carl Roth, Germany) containing 4% paraformaldehyde for 30–40 min. Brains were postfixed in 4% paraformaldehyde for 24 hr.

## Behavioral testing

### Examination of developmental milestones

Mouse pups were tested for their somatic development and reflexes at P2, P5 and P8. Weight, body and tail length were assessed. Surface righting reflex was quantified as time (max 30 s) until the pup turned over with all four feet on the ground after being placed on its back. Cliff aversion reflex was quantified as time (max 30 s) until the pup withdrew after snout and forepaws were positioned over an elevated edge. Vibrissa placing was rated positive if the pup turned its head after gently touching the whiskers with a toothpick.

## Electrophysiology

### Electrophysiological recording

A one-shank electrode (NeuroNexus, MI, USA) containing 1 × 16 recording sites (0.4–0.8 MΩ impedance, 100 mm spacing) was inserted into the layer V/VI of PFC. One-shank optoelectrodes (NeuroNexus, MI, USA) containing 1 × 16 recordings sites (0.4–0.8 MΩ impedance, 50 mm spacing) aligned with an optical fiber (105 mm diameter) ending 200 µm above the top recording site was inserted into either dHP or i/vHP. A silver wire was inserted into the cerebellum and served as ground and reference electrode. A recovery period of 10 min following insertion of electrodes before acquisition of data was provided. Extracellular signals were band-pass filtered (0.1–9,000 Hz) and digitized (32 kHz) with a multichannel extracellular amplifier (Digital Lynx SX; Neuralynx, Bozeman, MO, USA) and the Cheetah acquisition software (Neuralynx). Spontaneous (i.e. not induced by light stimulation) activity was recorded for 15 min at the beginning and end of each recording session as baseline activity. Only the baseline prior to stimulation epochs was used for data analysis. The position of recording electrodes in PL and CA1 area of dHP or i/vHP was confirmed after histological assessment post-mortem. For the analysis of prelimbic LFP, the recording site centered in PL was used, whereas for the analysis of spiking activity two channels above and two channels below this site were additionally considered. Recording site in cingulate or infralimbic sub-divisions of the PL were excluded from analysis. For the analysis of hippocampal LFP, the recording site located in str pyr,where sharp-waves reverse (*Bitzenhofer and Hanganu-Opatz, 2014*), was used to minimize any non-stationary effects of the large amplitude events. For the analysis of hippocampal firing, two channels below and two channels above this site were additionally considered.

## Light stimulation

Pulsed (laser on-off) light or ramp (linearly increasing power) stimulations were performed with an arduino uno (Arduino, Italy) controlled diode laser (473 nm or 600 nm; Omicron, Austria). Laser power was adjusted to trigger neuronal spiking in response to >25% of 3-ms-long light pulses at 16 Hz. Resulting light power was in the range of 20–40 mW/mm$^2$ at the fiber tip. For each frequency used (4, 8 and 16 Hz), stimuli (3 ms pulse length, 3 s stimulation duration, 6 s inter stimulation interval) were repeated (30 times) in a randomized order.

## Histology

### Immunohistochemistry

Brains were sectioned coronally at 50 μm. Free-floating slices were permeabilized and blocked with PBS containing 0.2% Triton X 100 (Sigma-Aldrich, MO, USA), 10% normal bovine serum (Jackson Immuno Research, PA, USA) and 0.02% sodium azide. Subsequently, slices were incubated overnight with mouse monoclonal Alexa Fluor-488 conjugated antibody against NeuN (1:200, MAB377X, Merck Millipore, MA, USA) or rabbit polyclonal primary antibody against GABA (1:1,000, A2052; Sigma-Aldrich), followed by 2 hr incubation with Alexa Fluor-488 goat anti-rabbit IgG secondary antibody (1:500, A11008; Merck Millipore). Slices were transferred to glass slides and covered with Fluoromount (Sigma-Aldrich, MO, USA).

For 3.3'-diaminobenzidie (DAB) staining sections (prepared as described above) were rinsed in PBS (0.125 M, pH 7.4–7.6) for 10 min, treated with peroxide solution (3% peroxide, 10% methanol in 0.125 M PB) for 10 min to quench any endogenous peroxidases within the tissue, and rinsed again. Subsequently, the sections were washed in PBS containing 0.5% Triton-X and incubated with avidin biotinylated enzyme complex (ABC, VECTASTAIN ABC Kit, USA) at room temperature or overnight at 4°C. After rinsing in Tris-HCl (pH 7.4), the sections were further incubated with DAB working buffer (DAB peroxidase substrate kit, Vector Laboratories, USA) at room temperature for 2–10 min. After the signal was detected, all sections were rinsed with Tris-HCl.

### Imaging

Wide field fluorescence was performed to reconstruct the recording electrode position in brain slices of electrophysiologically investigated pups and to localize tDimer2 expression in pups after IUE. High magnification images were acquired with a confocal microscope (DM IRBE, Leica, Germany) to quantify tDimer2 expression and immunopositive cells (1–4 brain slices/investigated mouse). For DAB staining, brightfield images were obtained using Zeiss imager M1 microscope (Zeiss, Oberkochen, Germany) and enhanced using the National Institutes of Health (NIH) Image program.

## Quantification and statistical analysis

### Immunohistochemistry quantification

All images were similarly analyzed with ImageJ. For quantification of fluorogold tracing automatic cell counting was done using custom-written tools. To quantify tDimer2, NeuN and GABA-positive neurons, manual counting was performed, since the high neuronal density in str. pyr. prevented reliable automatic counting.

### Spectral analysis of LFP

Data were imported and analyzed offline using custom-written tools in the Matlab environment (MathWorks). Data were processed as follows: band-pass filtered (500–5,000 Hz) to analyze MUA and low-pass filtered (<1,400 Hz) using a third-order Butterworth filter before downsampling to 3.2 kHz to analyze LFP. All filtering procedures were performed in a manner preserving phase information.

### Detection of oscillatory activity

The detection and of discontinuous patterns of activity in the neonatal PL and HP were performed using a modified version of the previously developed algorithm for unsupervised analysis of neonatal oscillations (*Cichon et al., 2014*) and confirmed by visual inspection. Briefly, deflections of the root mean square of band-pass filtered signals (1–100 Hz) exceeding a variance-depending threshold

were assigned as network oscillations. The threshold was determined by a Gaussian fit to the values ranging from 0 to the global maximum of the root-mean-square histogram. If two oscillations occurred within 200 ms of each other, they were considered as one. Only oscillations lasting >1 s was included.

## Detection of sharpwaves

Sharpwaves were detected by subtracting the filtered signal (1–300 Hz) from the recording sites 100 μm above and 100 μm below the recording site in str. pyr. Sharpwaves were then detected as peaks above five times the standard deviation of the subtracted signal.

## Power spectral density

Power spectral density was calculated using the Welch's method. Briefly, segments of the recorded signal were glued together (1 s segments for oscillatory activity; 300 ms segments for sharpwave pre/post comparison; 100 ms segments for ripple comparison; 3 s for light evoked activity) and power were then calculated using non-overlapping windows. Time–frequency plots were calculated by transforming the data using Morlet continuous wavelet.

## Coherence

Coherence was calculated using the imaginary coherency method (*Nolte et al., 2004*). Briefly, the imaginary coherence was calculated by taking the imaginary component of the cross-spectral density between the two signals and normalized by the power spectral density of each. The computation of the imaginary coherence C over frequency (f) for the power spectral density P of signal X and Y was performed according to the formula:

$$C_{XY}(f) = Im\left(\frac{|P_{XY}(f)|^2}{P_{XX}(f)P_{YY}(f)}\right)$$

## General partial directed coherence

gPDC is based on linear Granger causality measure. The method attempts to describe the causal relationship between multivariate time series based on the decomposition of multivariate partial coherences computed from multivariate autoregressive models. The LFP signal was divided into segments containing the oscillatory activity. Signal was de-noised using wavelets with the Matlab wavelet toolbox. After de-noising, gPDC was calculated using the gPDC algorithm previously described (*Baccala et al., 2007*).

## Single unit activity analysis

SUA was detected and clustered using Offline Sorter (Plexon, TC, USA). 1–4 single units were detected at each recording site. Subsequently, data were imported and analyzed using custom-written tools in the Matlab software (MathWorks). The firing rate temporally related to SPWs was calculated by aligning all units to the detected SPWs. For assessing the phase locking of units to LFP, we firstly used the Rayleigh test for non-uniformity of circular data to identify the units significantly locked to network oscillations. The phase was calculated by extracting the phase component using the Hilbert transform of the filtered signal at each detected spike. Spikes occurring in a 15 ms-long time window after the start of a light pulse were considered to be light-evoked. Stimulation efficacy was calculated as the probability of at least one spike occurring in this period.

## Statistical analysis

Statistical analyses were performed using SPSS Statistics 21 (IBM, NY, USA) or Matlab. Data were tested for normal distribution by the Shapiro–Wilk test. Normally distributed data were tested for significant differences (*p<0.05, **p<0.01 and ***p<0.001) using paired t-test, unpaired t-test or one-way repeated-measures analysis of variance with Bonferroni-corrected post hoc analysis. Not normally distributed data were tested with the nonparametric Mann–Whitney U-test. The circular statistics toolbox was used to test for significant differences in the phase locking data. Data are presented as mean ±SEM. No statistical measures were used to estimate sample size since effect size was unknown. Investigators were not blinded to the group allocation during the experiments.

Unsupervised analysis software was used if possible to preclude investigator biases. Summary of performed statistical analysis is summarized in *Supplementary file 1*.

## Acknowledgements

We thank Nadine Faesel for the establishment of tracing technique, Antonio Candela for preliminary tracing data, Drs. L Cancedda and A Cwetsch for help with the *in utero* electroporation, Drs. S Wiegert, T Oertner, and C Gee for help with the development of opsin constructs as well as A Marquardt, P Putthoff, A Dahlmann and I Ohmert for excellent technical assistance. ILH-O acknowledges support by the ERC (ERC Consolidator Grant 681577) and by the German Research Foundation (SFB 936 (B5) and SPP 1665 (Ha4466/12-1)). ILH-O is the member of the FENS Kavli Network of Excellence. In memoriam of Howard Eichenbaum

## Additional information

### Funding

| Funder | Grant reference number | Author |
|---|---|---|
| European Research Council | 681577 | Ileana L Hanganu-Opatz |
| Deutsche Forschungsgemeinschaft | SFB 936 | Ileana L Hanganu-Opatz |
| Deutsche Forschungsgemeinschaft | SPP 1665 | Ileana L Hanganu-Opatz |

The funders had no role in study design, data collection and interpretation, or the decision to submit the work for publication.

### Author contributions

Joachim Ahlbeck, Data curation, Formal analysis, Investigation, Methodology, Writing—original draft, Writing—review and editing; Lingzhen Song, Data curation, Formal analysis, Writing—review and editing; Mattia Chini, Data curation, Formal analysis, Investigation, Methodology, Writing—review and editing; Sebastian H Bitzenhofer, Data curation, Validation, Investigation, Visualization, Methodology, Writing—review and editing; Ileana L Hanganu-Opatz, Conceptualization, Data curation, Supervision, Funding acquisition, Validation, Investigation, Methodology, Writing—original draft, Project administration, Writing—review and editing

### Author ORCIDs

Joachim Ahlbeck (iD) https://orcid.org/0000-0002-4439-0798
Mattia Chini (iD) http://orcid.org/0000-0002-5782-9720
Ileana L Hanganu-Opatz (iD) http://orcid.org/0000-0002-4787-1765

### Ethics

Animal experimentation: All experiments were performed in compliance with the German laws and the guidelines of the European Community for the use of animals in research and were approved by the local ethical committee (111/12, 132/12).

### Decision letter and Author response

Decision letter https://doi.org/10.7554/eLife.33158.022
Author response https://doi.org/10.7554/eLife.33158.023

# Additional files

## Supplementary files

• Supplementary file 1. (table supplement 1 for *Figures 1–5* and supplementary figures 1-4) Summary of statistics for all experiments. (A) Statistical testing, number of investigated mice and p-values for the analyses displayed in *Figure 1*. (B)–(I) Same as (A) for analyses in *Figures 2–6*, S1-4.
DOI: https://doi.org/10.7554/eLife.33158.016

• Source code 1. Matlab source code for the analysis of discontinuous oscillatory activity.
DOI: https://doi.org/10.7554/eLife.33158.017

• Transparent reporting form
DOI: https://doi.org/10.7554/eLife.33158.018

## Major datasets

The following dataset was generated:

| Author(s) | Year | Dataset title | Dataset URL | Database, license, and accessibility information |
| --- | --- | --- | --- | --- |
| Ahlbeck J, Song L, Chini M, Candela A, Bitzenhofer S, Hanganu-Opatz I | 2018 | Data from: Glutamatergic drive along the septo-temporal axis of hippocampus boosts prelimbic oscillations in the neonatal mouse | http://dx.doi.org/10.5061/dryad.52fh | Available at Dryad Digital Repository under a CC0 Public Domain Dedication |

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
