## [Decision Letter]

Thank you for submitting your article "Glutamatergic drive along the septo-temporal axis of hippocampus boosts prelimbic oscillations in the neonatal mouse" for consideration by *eLife*. Your article has been favorably evaluated by Gary Westbrook (Senior Editor) and three reviewers, one of whom is a member of our Board of Reviewing Editors. The following individual involved in review of your submission has agreed to reveal his identity: Jonas-Frederic Sauer (Reviewer #3). The reviewers have discussed the reviews with one another and the Reviewing Editor has drafted this decision to help you prepare a revised submission.

Summary:

The study by Ahlbeck et al. examines the early postnatal development of long-range projections between the hippocampus and the prelimbic cortex, particularly their role in driving oscillatory activity patterns in the prelimbic cortex of young mice. In this study the authors apply a broad set of techniques including electrophysiological recordings of local field potentials as well as single units, axonal tracing and optogenetic applications to examine first, what activity patterns are expressed in the developing dorsal and ventral hippocampus and second, whether glutamatergic drive originating in the hippocampus influences the activity in the prelimbic area. The study shows that the ventral hippocampus is anatomically strongly coupled to the prelimbic area and drives theta- and β/γ-modulated activity patterns in the prelimbic cortex. In contrast, the dorsal hippocampus shows locally higher theta power but is less involved in this drive.

Essential revisions:

1) All reviewers agreed that this is a very interesting study that nicely builds upon the authors previous work and provides strong evidence for a topographic drive for prefrontal activity located in the v/i portion of the neonatal hippocampus. However, one of the main criticisms was that the current set of experiments does not directly prove the hypothesis that 'theta in the HP drives the oscillatory entrainment of the local circuits in the PFC'. It is not entirely clear whether the number of optogenetically activated HP pyramidal cells reflects the number of cells active during a natural oscillation event. Furthermore, during optogenetic stimulation, the spikes might be more tightly synchronized compared to naturally occurring oscillations, which might favor downstream synchronization.

Rather, the manuscript shows that driving a substantial proportion of HP pyramidal neurons at theta frequency induces PFC oscillations. Thus, a causal proof requires additional experiments, such as specific silencing of the ventral hippocampus (optogenetically or pharmacologically). Without this experiment the relationships remain only correlational or modulatory.

2) More details are required on where from the LFP recordings were made since the amplitude and waveforms are strongly dependent on cortical and hippocampal depth. The authors should perform CSD analysis to analyze the amplitude and the power of theta and ripple oscillations both in the hippocampus and the prefrontal cortex.

3) Please provide averaged single optostimulus – triggered LFP/unit PETH in the PFC – as functional evidence for the connections.

4) One of the emerging questions from the study is whether periodic ontogenetic activation of ventral hippocampal principal cells has a similar effect on the activity patterns in the prelimbic area than in the adult mice. Moreover, it remains unclear why the 8 Hz ontogenetic stimulation of hippocampal principal cells is more efficient than 4 and/or 16 Hz stimulation in driving prefrontal activity patterns? These questions can be addressed in the Discussion section of the revised manuscript.

[Editors' note: further revisions were requested prior to acceptance, as described below.]

Thank you for resubmitting your work entitled "Glutamatergic drive along the septo-temporal axis of hippocampus boosts prelimbic oscillations in the neonatal mouse" for further consideration at *eLife*. Your revised article has been favorably evaluated by Gary Westbrook (Senior Editor), a Reviewing Editor, and two reviewers.

The figures and the main body of the text have been improved according to the comments, but there are some remaining issues that need to be addressed before acceptance, as outlined below:

1) Error in the text, subsection “Theta activity within dorsal and intermediate / ventral hippocampus differently entrains the neonatal prelimbic cortex”, second paragraph: Figure 1—figure supplement 2D and not Figure 2—figure supplement 2D.

2) There seems to be a confusion about the numbering of the Figure supplements. The included references to Figure 2—figure supplement 2D (subsection “Theta activity within dorsal and intermediate / ventral hippocampus differently entrains the neonatal prelimbic cortex”, second paragraph) and Figure 3—figure supplement 2E (subsection “SPWs-mediated output of intermediate / ventral but not dorsal hippocampus times network oscillations and spiking response in the neonatal prelimbic cortex”, first paragraph) both point to Figure 1—figure supplement 2. To meet eLife’s guidelines on figures that should be linked to a main figure, I would suggest to place the content of Figure 1—figure supplement 2D in a new Figure 2—figure supplement 1, and Figure 1—figure supplement 2E in a new Figure 3—figure supplement 1.

3) References to 'Figure S2A' and 'Figure S2B' in the subsection “Neonatal dorsal and intermediate / ventral hippocampus are differently entrained in discontinuous patterns of oscillatory activity” should be corrected to 'Figure 1—figure supplement 2A' and 2B', respectively.

4) Detailed and rigorous statistical information should be supplied in an excel table.

---

## [Author Response]

Essential revisions:1) All reviewers agreed that this is a very interesting study that nicely builds upon the authors previous work and provides strong evidence for a topographic drive for prefrontal activity located in the v/i portion of the neonatal hippocampus. However, one of the main criticisms was that the current set of experiments does not directly prove the hypothesis that 'theta in the HP drives the oscillatory entrainment of the local circuits in the PFC'. It is not entirely clear whether the number of optogenetically activated HP pyramidal cells reflects the number of cells active during a natural oscillation event. Furthermore, during optogenetic stimulation, the spikes might be more tightly synchronized compared to naturally occurring oscillations, which might favor downstream synchronization.Rather, the manuscript shows that driving a substantial proportion of HP pyramidal neurons at theta frequency induces PFC oscillations. Thus, a causal proof requires additional experiments, such as specific silencing of the ventral hippocampus (optogenetically or pharmacologically). Without this experiment the relationships remain only correlational or modulatory.

We performed additional experiments to reinforce the causal relationship between hippocampal and prefrontal activity. For this, we modulated the hippocampal activity by selectively silencing or activating the interneurons transfected with excitatory (ChETA) or inhibitory (ArchT) light-sensitive proteins with blue and yellow light, respectively. Silencing the interneuronal activity in HP caused prominent oscillatory discharges both in HP and PFC, whereas its activation diminished the oscillatory entrainment within prefrontal-hippocampal circuits. We added the new data to Results (subsection “Selective light manipulation activation of pyramidal neurons and interneurons in CA1 area of intermediate / ventral but not dorsal hippocampus causes frequency-specific changes in the oscillatory entrainment of neonatal prelimbic circuits”, fifth paragraph) and supplemented the manuscript with a new main figure (Figure 6). These findings complement our previously published results when pharmacological silencing of hippocampus led to diminishment of prefrontal oscillatory activity (Brockmann et al., 2011).

2) More details are required on where from the LFP recordings were made since the amplitude and waveforms are strongly dependent on cortical and hippocampal depth. The authors should perform CSD analysis to analyze the amplitude and the power of theta and ripple oscillations both in the hippocampus and the prefrontal cortex.

We characterized the spatial organization of prefrontal activity in previous studies (Cichon et al., 2014) when using multi-shank electrodes spanning all layers of prelimbic cortex. To avoid redundancy, we did not include similar findings in the present paper. The exact location of recording electrodes in PFC and HP was confirmed after histological investigation post-mortem. We supplemented the manuscript with details on the selection of recording sites for LFP and spiking analysis (Materials and methods, subsection “Electrophysiology”). Moreover, we illustrated the CSD analysis of recordings in dHP and i/vHP in Figure 1—figure supplement 2F.

3) Please provide averaged single optostimulus – triggered LFP/unit PETH in the PFC – as functional evidence for the connections.

We provided the requested information by adding an inset to Figure 5C.

4) One of the emerging questions from the study is whether periodic ontogenetic activation of ventral hippocampal principal cells has a similar effect on the activity patterns in the prelimbic area than in the adult mice. Moreover, it remains unclear why the 8 Hz ontogenetic stimulation of hippocampal principal cells is more efficient than 4 and/or 16 Hz stimulation in driving prefrontal activity patterns? These questions can be addressed in the Discussion section of the revised manuscript.

We addressed these issues in Discussion (subsection “Optogenetic interrogation of long-range coupling in the developing brain”, second paragraph).

[Editors' note: further revisions were requested prior to acceptance, as described below.]

The figures and the main body of the text have been improved according to the comments, but there are some remaining issues that need to be addressed before acceptance, as outlined below:1) Error in the text, subsection “Theta activity within dorsal and intermediate / ventral hippocampus differently entrains the neonatal prelimbic cortex”, second paragraph: Figure 1—figure supplement 2D and not Figure 2—figure supplement 2D.

Corrected.

2) There seems to be a confusion about the numbering of the Figure supplements. The included references to Figure 2—figure supplement 2D (subsection “Theta activity within dorsal and intermediate / ventral hippocampus differently entrains the neonatal prelimbic cortex”, second paragraph) and Figure 3—figure supplement 2E (subsection “SPWs-mediated output of intermediate / ventral but not dorsal hippocampus times network oscillations and spiking response in the neonatal prelimbic cortex”, first paragraph) both point to Figure 1—figure supplement 2. To meet eLife’s guidelines on figures that should be linked to a main figure, I would suggest to place the content of Figure 1—figure supplement 2D in a new Figure 2—figure supplement 1, and Figure 1—figure supplement 2E in a new Figure 3—figure supplement 1.

We modified to meet the *eLife* guidelines.

3) References to 'Figure S2A' and 'Figure S2B' in the subsection “Neonatal dorsal and intermediate / ventral hippocampus are differently entrained in discontinuous patterns of oscillatory activity” should be corrected to 'Figure 1—figure supplement 2A' and 2B', respectively.

Corrected.

4) Detailed and rigorous statistical information should be supplied in an excel table.

We added the statistics table.